# Cholesterol Biosynthesis Modulates CSFV Replication

**DOI:** 10.3390/v14071450

**Published:** 2022-06-30

**Authors:** Xiaodong Zou, Feng Lin, Yang Yang, Jiahuan Chen, Huanyu Zhang, Linquan Li, Hongsheng Ouyang, Daxin Pang, Xiaochun Tang

**Affiliations:** 1College of Animal Sciences, Jilin University, Changchun 130062, China; zouxd19@mails.jlu.edu.cn (X.Z.); linfeng21@mails.jlu.edu.cn (F.L.); y_y21@mails.jlu.edu.cn (Y.Y.); chensiyi@jlu.edu.cn (J.C.); zhy20@mails.jlu.edu.cn (H.Z.); lilq20@mails.jlu.edu.cn (L.L.); ouyh@jlu.edu.cn (H.O.); pdx@jlu.edu.cn (D.P.); 2Chongqing Research Institute, Jilin University, Chongqing 401120, China

**Keywords:** classical swine fever virus (CSFV), lipid metabolism, cholesterol biosynthesis

## Abstract

Classical swine fever (CSF) caused by the classical swine fever virus (CSFV) has resulted in severe losses to the pig industry worldwide. It has been proposed that lipid synthesis is essential for viral replication, and lipids are involved in viral protein maturation and envelope production. However, the specific crosstalk between CSFV and host cell lipid metabolism is still unknown. In this study, we found that CSFV infection increased intracellular cholesterol levels in PK-15 cells. Further analysis demonstrated that CSFV infection upregulated PCSK9 expression to block the uptake of exogenous cholesterol by LDLR and enhanced the cholesterol biosynthesis pathway, which disrupted the type I IFN response in PK-15 cells. Our findings provide new insight into the mechanisms underpinning the pathogenesis of CSFV and hint at methods for controlling the disease.

## 1. Introduction

Classical swine fever (CSF) is a highly contagious, economically significant, multisystemic viral disease in swine that has bad adverse effects on the pig industry worldwide [1]. The incubation period of CSF varies between 3 and 10 days, which is influenced by viral virulence and the age of infected animals. CSF is characterized by several clinicopathological signs, including high fever, constipation and/or diarrhea, and skin hemorrhages of infected animals [2]. Infected pregnant sows manifest abortions, stillbirths, mummified fetuses, and malformations [3]. As a causative agent, the classical swine fever virus (CSFV) is an enveloped, positive-sense, single-stranded RNA virus belonging to the Pestivirus genus in the Flaviviridae family. The CSFV genome possesses a single open reading frame (ORF) with a flanked non-translated region at both the end of the genome (5′- and 3′-NCRs). It encodes a polypeptide precursor of 3898 amino acids (aa) that is post-translationally cleaved into four structural proteins (C, Erns, E1, and E2) and eight non-structural proteins (Npro, p7, NS2, NS3, NS4A, NS4B, NS5A, and NS5B) by viral and cellular proteases [4].

Viruses are small intracellular parasites relying on cellular factors for replication, and the fundamental aspects of CSFV–host interactions have been well described [5]. Cytoplasmic RNA helicase A (RHA) is involved in the modulation of RNA synthesis, replication, and translation of CSFV via binding 5′- and 3′-NCRs of CSFV [6]. CSFV Npro interacts with interferon regulatory factor 3 (IRF3), IRF7, and poly(C)-binding protein 1(PCBP1) to block interferons’ (IFNs) production for viral replication [7,8,9]. In addition, CSFV Npro binds to HS-1-associated protein X-1 (HAX-1) to enhance cellular resistance to apoptosis [10]. CSFV core protein interacts with SUMO-conjugating enzyme (UBC9), small ubiquitin-like modifier (SUMO-1), and IQ motif containing GTPase activating protein 1 (IQGAP1) to maintain viral virulence [11,12]. Moreover, the interaction between CSFV core protein and osteosarcoma amplified protein 9 (OS9) inhibits the virus replication in the cell culture [13]. CSFV E2 protein interacts with cellular β-actin during the early stage of the replication cycle by affecting the intracellular transport process of CSFV or E2 protein in the cell at the post-entry step [14]. In CSFV-infected PK-15 cells, annexin a2 (Anx2) is upregulated and interacts with E2 protein to promote CSFV replication [15]. Anx2 also binds to CSFV NS5A and enhances virus assembly [16]. Moreover, CSFV NS5A protein interacts with host factors such as heat shock protein 70 (HSP70) and guanylate-binding protein 1 (GBP1) to regulate viral replication [17].

Nowadays, perturbation of lipid metabolism homeostasis emerges as an important feature during viral infections, and viruses use host lipid machinery to support their replication cycle and impair the host immune response [18]. Human cytomegalovirus (HCMV) uses a host stress response to balance the elongation of saturated/monounsaturated and polyunsaturated very-long-chain fatty acids [19]. In addition, HCMV and Middle East respiratory syndrome coronavirus (MERS-CoV) increase SREBP1 translocation to the cell nucleus for enhancing fatty acid synthesis [20,21]. Some viruses, such as the Epstein–Barr virus (EBV) and respiratory syncytial virus (RSV), upregulate the expression of fatty acid synthase (FASN) and promote lipogenic gene expression [22,23,24,25]. Other viruses, such as severe acute respiratory syndrome coronavirus 2 (SARS-CoV-2), hepatitis c virus (HCV), and zika virus (ZIKV), affect the trafficking, formation, and recruitment processes of cellular lipid droplets (LDs) [26,27,28,29,30,31,32]. Moreover, glutathione peroxidase 4 (GPX4) inactivation increases the production of lipid peroxidation, leading to STING carbonylation at C88, suppressing its trafficking from the endoplasmic reticulum (ER) to the Golgi complex, blocking the innate antiviral immune responses, and promoting HSV-1 replication in vivo [33]. During SARS-CoV-2 biogenesis in infected cells, the spike protein is rapidly and efficiently S-acylated via the sequential action of the ZDHHC20 and ZDHHC9 enzymes to control SARS-CoV-2 membrane lipid organization and enhance infectivity [34].

Several lines of evidence suggest the crucial role of lipid metabolism in CSFV replication. Serum lipidomic analysis of piglets infected with CSFV indicated that CSFV required free fatty acids to enhance viral replication [35]. Moreover, CSFV infection upregulates the expression of FASN and fatty acid synthesis for lipid accumulation to maintain viral replication [36,37]. However, the exact function of cellular lipid metabolism in the CSFV life cycle remains unclear.

Our study explored the crosstalk between CSFV infection and cholesterol metabolism in PK-15 cells and the underlying mechanisms. Our results showed that CSFV infection increased cellular cholesterol levels and cholesterol biosynthesis modulated CSFV replication. Moreover, PCSK9 protein expression was increased to counteract exogenous cholesterol uptake during CSFV infection. Our findings provide new insights for future anti-CSFV research, and cholesterol biosynthesis can be exploited as a potential target.

## 2. Materials and Methods

### 2.1. Cells and Virus

Porcine kidney cell line (PK-15) cells (ATCC, CCL-33) were cultured in DMEM (Gibco, Waltham, MA, USA) supplemented with 5% fetal bovine serum (FBS, thermally inactivated at 56 °C for 30 min) at 37 °C and 5% CO_2_ in a humidified incubator. Dr. Changchun Tu kindly provided the CSFV Shimen strain (Academy of Military Medical Sciences, Changchun, China).

### 2.2. Quantitative Real-Time PCR (qPCR)

Total cellular RNAs were extracted using TRNzol-A+ Reagent (Tiangen, Beijing, China) and transcribed into cDNA using FastKing RT Kit (with gDNase) (Tiangen, China) according to the manufacturer’s instructions. We then used the iCycler Thermal Cycler with iQ5 Optical Module for RT-PCR to measure fluorescence intensity and amplification plots (Bio-Rad, Hercules, CA, USA, ABI 7500, iQ5). The primers used in qPCR are listed in Appendix A.

### 2.3. Virus One-Step Growth Curve

PK-15 cells were seeded into 6-well plates and infected with 200 TCID_50_ of CSFV for 1 h, washed with PBS three times, and cultured in a fresh medium. The viral genome copies were detected at 12, 24, 36, 48, 60, 72, 84, and 96 h post-infection (hpi). A total of 200 μL of viral supernatant was harvested, viral RNA extracted, and reverse transcribed into cDNA. Then, viral genome copies of CSFV were quantified by qPCR, and the one-step growth curve was plotted according to viral genome copies.

### 2.4. Quantification of Cholesterol Contents

Cells were lysed and incubated at 4 °C for 30 min; then, the supernatants were collected and clarified by centrifugation at 3000 rpm. The amount of cholesterol in the samples was measured using the total cholesterol assay kit (Nanjing Jiancheng Bioengineering Institute, Nanjing, China) according to the manufacturer’s instructions. The cholesterol level was measured using an infinite 2000 PRO Microplate Reader (Tecan, Männedorf, Switzerland), and three applications monitored the absorbance of the samples. The results were normalized by the total protein concentration of each sample.

### 2.5. Immunofluorescence Assay (IFA)

PK-15 cells were fixed in 4% paraformaldehyde for 30 min at room temperature, washed with PBS three times, permeabilized in 0.25% Triton X-100 for 15 min, and blocked with normal goat serum for 30 min at room temperature. The cells were then incubated with mouse monoclonal antibody (mAb) WH303 specific to CSFV E2 (Bioss, China) overnight at 4 °C, followed by incubation with goat anti-rabbit IgG H&L (Alexa Fluor^®^ 488) secondary antibody (Abcam, Boston, MA, USA) for 2 h at 37 °C, and washed with PBS three times. Finally, cell nuclei were stained with Hoechst 33,342 (Dojindo, Shanghai, China) for 10 min and observed under a fluorescence microscope.

### 2.6. Transcriptome Analysis

Total RNA was extracted from each group, and the construction of the RNA-seq library and sequencing of the libraries was performed by a commercial service. The Bioconductor package edgeR was used to perform differential expression analysis of the different libraries. Pathway enrichment analyses were conducted for the significant expression genes identified via the Metascape platform (http://metascape.org/, accessed on 20 April 2022) [38].

### 2.7. Western Blotting (WB)

PK-15 cells were lysed in cell lysis buffer containing 1 mM PMSF and incubated at 4 °C for 2 h. The supernatants were collected and clarified by centrifugation at 12,000 rpm for 30 min. Then, the protein concentration was quantified via a BCA assay kit (Beyotime, China) according to the manufacturer’s instructions. Equal amounts of proteins were diluted with 5× loading buffer, boiled for 10 min, separated on the SDS-PAGE, and electrophoretically transferred to PVDF membranes. The membranes were blocked with 5% skimmed milk in PBST, incubated with antibodies in PBST, and scanned by a Tanon-5200 Chemiluminescent Imaging System (Tanon Science and Technology, Shanghai, China). The data were analyzed by Image J software, and the primary and secondary antibodies involved in the process are shown in Appendix A.

### 2.8. LDL Uptake Assay

PK-15 cells were seeded into 24-well plates with 30~40% confluence and treated with a serum starvation medium for 12 h. The cells were washed with DPBS containing BSA and treated with 500 μg/mL of heparin for 1 h. Then, the cells were washed and incubated with 10 μg/mL of dil-LDL solution for 4 h at 37 °C. The cells were removed from the 37 °C incubation and rinsed with wash buffer at room temperature. Subsequently, the cell nuclei were stained with Hoechst 33,342 (Dojindo, China) for 10 min at 37 °C. Finally, the cells were fixed in 4% paraformaldehyde for 30 min at room temperature and analyzed with a fluorescent microscope.

### 2.9. Nile Red Staining

PK-15 cells were seeded into 12-well plates with 80~90% confluence and fixed in 4% paraformaldehyde for 30 min. Next, the cells were incubated with Nile red solution for 10 min at 37 °C and washed with PBS three times. Then, cell nuclei were stained with Hoechst 33,342 (Dojindo, China) for 10 min at 37 °C and observed under a fluorescent microscope.

### 2.10. Construction of Plasmid

CRISPR/Cas9-single-guide RNAs targeting *LDLR*, *SQLE,* and *HSD17B7* were designed by the online software (http://crispr.mit.edu/, accessed on 20 April 2022), synthesized (Comate Bioscience, Changchun, China), annealed, and cloned into the pX330-U6-Chimeric BB-CBh-hSpCas9 empty expression vector (Addgene, 42230) or the pBluescriptSKII+U6-sgRNA(F+E) empty expression vector (Addgene, 74707).

### 2.11. Transfection and Genotyping of Cell Clones

PK-15 cells were electrotransfected with 30 μg plasmids in 200 μL of Opti-MEM (GIBCO, Grand Island, NY, USA) using 2 mm gap cuvettes and a BTX ECM 2001 electroporator according to the manufacturer’s instructions. After electrotransfection, the cells were cultured at 37 °C for 36 h and plated into 10-cm dishes for 7–8 days. Then, the single-cell colonies were picked and placed with amplification culture in 48-well plates. After growing to 70–80% confluence, the cell clones were digested by trypsin, and 50% of each clone was lysed by NP40 lysis buffer containing 0.45% NP-40 plus 0.6% Proteinase K. The lysate was used as the temple for genotyping-PCR amplification, and the primers are shown in the following Table 1.

### 2.12. Statistical Analysis

All data are expressed as the means ± standard error of the mean (SEM). The unpaired Student’s *t*-test determined statistical differences for two-group comparisons and one-way ANOVA with Bonferroni’s post-test for multiple group comparisons with a *p*-value < 0.05. All statistical analyses were completed using GraphPad Prism 7.0 software.

## 3. Results

### 3.1. CSFV Upregulates Intracellular Cholesterol Levels in PK-15 Cells

A one-step growth curve of CSFV was detected, and the results showed that the copy numbers of viral genomic RNAs peaked 48 h post-infection (hpi) (Figure 1a). Then, the intracellular cholesterol levels were examined during CSFV infection, and the results showed that intracellular cholesterol levels significantly increased in CSFV-infected PK-15 cells (Figure 1b). Further analysis showed that the correlation coefficient between CSFV replication and intracellular cholesterol contents was 0.9142, and the *p*-value was 0.0015, indicating that viral replication was significantly positively correlated with intracellular cholesterol contents (Figure 1c). Furthermore, Nile red staining of cholesterol in CSFV-infected PK-15 cells at 48 hpi showed that the intracellular cholesterol levels increased in CSFV-infected PK-15 cells. CSFV infection significantly increased intracellular cholesterol levels in PK-15 cells, and the two positively correlated.

### 3.2. CSFV Enhances Cholesterol Biosynthesis in PK-15 Cells

To further explore the effects of CSFV on intracellular cholesterol levels, CSFV-infected PK-15 cells were prepared at 48 hpi for transcriptome analysis. The analysis results of gene ontology (GO) and the Kyoto encyclopedia of genes and genomes (KEGG) pathway enrichment showed that the “cholesterol biosynthesis pathway” was a highly enriched biological pathway in CSFV-infected PK-15 cells (Figure 2a). Moreover, the transcriptional expression analysis confirmed significant changes in genes involved in cholesterol biosynthesis in RNA-seq (Figure 2b,c). Collectively, CSFV infection altered host transcriptome levels and enhanced cholesterol biosynthesis in PK-15 cells.

### 3.3. The Uptake of Exogenous Cholesterol Was Blocked by PCSK9 in CSFV Infected PK15 Cells

Intracellular cholesterol levels comprise de novo synthesis and extracellular uptake. The PK-15 cells’ ability to uptake exogenous cholesterol was detected. PK-15 cells normally uptake exogenous dil-LDL, and starvation increases the uptake of dil-LDL as shown in Appendix A. Therefore, the source of increased intracellular cholesterol in CSFV-infected PK-15 cells was further investigated. PK-15 cells were treated with different cholesterol types and simultaneously infected with CSFV for 48 h. The results showed that small-molecule cholesterol (CH) rather than LDL-C could increase CSFV replication and intracellular cholesterol levels (Figure 3a,b). The Western blotting results showed that CSFV infection did not affect the LDLR level but significantly increased PCSK9 expression (Figure 3c,d). Moreover, SBC-115076, a potent PCSK9 antagonist, increased the uptake of exogenous dil-LDL during CSFV infection (Figure 3e,f) but could not enhance viral replication in PK-15 cells (Figure 3g). Further research found that *LDLR* knockout (LO) did not affect CSFV replication and intracellular cholesterol contents in PK-15 cells during CSFV infection (Figure 3h–j and Appendix A). Altogether, these data suggested that CSFV inhibited exogenous cholesterol uptake by upregulating PCSK9 expression.

### 3.4. Inhibitions of Cholesterol Biosynthesis Impaired CSFV Replication

The aforementioned results showed that increased intracellular cholesterol was not derived from LDLR uptake; therefore, we further explored the role of enhanced cholesterol biosynthesis in CSFV replication. The CRISPR/Cas9 system destroyed the cholesterol biosynthesis pathway to elucidate its effect on viral replication (Appendix A). The results showed that CSFV replication was effectively reduced in SOLE knockout (SO) and HSD17B7 knockout (HO) cells, and the effect of *SQLE* gene deficiency on CSFV replication was more significant than *HSD17B7* gene deficiency (Figure 4a). Moreover, as shown in the results of the cholesterol assay, dil-LDL uptake, and Nile red staining, SO and HO cells maintained intracellular cholesterol homeostasis by enhancing exogenous uptake capacity when not infected with CSFV. However, in the case of CSFV infection, the intracellular cholesterol levels in SO and HO cells were significantly lower than in the control group (Figure 4b–f). Collectively, the disruption of cholesterol biosynthesis significantly decreased CSFV replication.

## 4. Discussion

In this study, we showed that CSFV infection increased intracellular cholesterol levels, and cholesterol biosynthesis played an important role in CSFV replication. Meanwhile, we demonstrated that increased intracellular cholesterol is derived from intracellular biosynthesis rather than exogenous uptake by LDLR.

Growing evidence indicates an intimate relationship between CSFV and the host cellular cholesterol metabolism. Studies have shown that cellular cholesterol is essential for CSFV infection, and cholesterol depletion significantly blocks virus internalization and thus inhibits CSFV infection at the initial stage [39]. Moreover, inhibiting coatomer protein I (COP I) function and Niemann-pick C1 (NPC1) expression, which impairs cholesterol transport and leads to cholesterol and virion accumulation in early endosomes, thus suppresses CSFV invasion and replication [40,41]. To our knowledge, our study is the first detailed report on alterations to intracellular cholesterol in PK-15 cells during CSFV infection. We found that intracellular cholesterol levels increased significantly and reached their highest level at 48 hpi in CSFV-infected PK-15 cells. We further analyzed the correlation between CSFV replication and intracellular cholesterol and found that viral replication was significantly positively correlated with intracellular cholesterol levels. Altogether, our findings confirmed the relationship between intracellular cholesterol and CSFV replication.

Intracellular cholesterol homeostasis is regulated dynamically by exogenous uptake and de novo biosynthesis. Acquisition of exogenous cholesterol relies on the interaction of cells with low-density lipoproteins (LDL); plasma LDL-C engages the ubiquitously expressed LDL receptor (LDLR), and the complex is internalized to allow cholesterol uptake [42]. Studies have demonstrated how HCV and vesicular stomatitis virus (VSV) utilize LDLR to enter the host cell and that LDLR-mediated internalization of LDL is vital for HCV replication [43,44,45]. Other studies have shown that enhanced cholesterol biosynthesis is involved in viral life cycles. Gamma-herpes virus and HCMV reprogram cholesterol biosynthesis and enhance cholesterol flux for optimal viral production [46,47]. The inhibition of cholesterol biosynthesis by statins significantly impairs the replication of HIV, HCV, dengue virus (DENV), influenza A virus (IAV), and coxsackievirus B3 (CVB3) [48,49,50,51,52]. In the present study, we investigated the source of elevated cholesterol and the involvement of LDLR in CSFV-infected PK-15 cells. We found that small molecule cholesterol promotes CSFV replication rather than macromolecule LDL-C. Small molecule cholesterol enters the host cell by free diffusion, whereas macromolecules (LDL-C) require the assistance of LDLR. Our findings showed that CSFV upregulated PCSK9 expression to block exogenous cholesterol uptake by LDLR, and the disrupted LDLR did not affect CSFV replication in PK-15 cells. Moreover, the inhibition of PCSK9 expression enhanced exogenous cholesterol uptake but did not increase CSFV replication. Macromolecules enter cells via active transport, which requires host cell mitochondria to provide energy. However, studies have shown that CSFV hijacks host energy metabolism by triggering mitophagy and reducing mitochondrial mass for replication [53]. Furthermore, the transcriptome results further confirmed that cholesterol biosynthesis was significantly enhanced in CSFV-infected cells, and the deletion of SQLE and HSD17B7, the key enzymes involved in cholesterol biosynthesis, significantly impaired CSFV replication. Cholesterol biosynthesis is the key source of elevated intracellular cholesterol and plays an important role in CSFV replication.

Cholesterol biosynthesis uses Acetyl-CoA as the primary building block, producing a series of intermediates such as isoprenoids and sterols from the mevalonate and lanosterol pathways, respectively [54]. Previous research reports showed that the availability of cholesterol intermediates, which serve as substrates for protein prenylation, was important for murine-gammaherpesvirus-68 (MHV68) replication [46]. The presence of the cholesterol precursor, desmosterol, appears to partially rescue HSV-1 entry in the absence of cellular cholesterol [55]. Moreover, the inhibition of lanosterol significantly decreases human rhinovirus [56]. In this study, we found that lower viral copies in *SQLE* knockout cells are comparable to *HSD17B7* knockout cells. SQLE catalyzes the oxygenation step in cholesterol synthesis and turns isoprenoid squalene into 2,3-(S)-oxo squalene before lanosterol formation; HSD17B7 contributes to the conversion of lanosterol to zymosterol. Thus, we speculated that lanosterol might be involved in CSFV infection and play a role in viral replication.

Viruses reprogram cholesterol biosynthesis for optimal replication, and the host cells utilize type I interferon (IFN) signaling to create a positive feedback loop to maintain homeostasis [57]. Thus, we clarified the crosstalk between cholesterol and IFNs during CSFV replication. Interferon-stimulated genes (ISGs) such as ccl2, cxcl10, and mx1 and the phosphorylated TBK1 (p-TBK1) levels in CSFV-infected PK-15 cells were significantly suppressed at 48 hpi, and the disruption of the cholesterol biosynthesis pathway facilitated the expression of p-TBK1 and ISGs, as shown in Appendix A. These data suggest that CSFV promotes cholesterol biosynthesis to attenuate type I IFN signaling for viral replication.

In conclusion, our observations highlight that CSFV infection induces the cholesterol metabolism reprogramming of host cells, which facilitates replication. Mechanistically, the upregulation of PCSK9 expression by CSFV infection blocks exogenous cholesterol uptake, thus enhancing cholesterol biosynthesis, suppressing type I IFN response, and facilitating CSFV replication. Understanding the importance of cholesterol biosynthesis in CSFV replication offers new insights for future anti-CSFV research. Further studies are warranted to elucidate the precise mechanisms between cholesterol biosynthesis and CSFV replication.

## Figures and Tables

**Figure 1 viruses-14-01450-f001:**
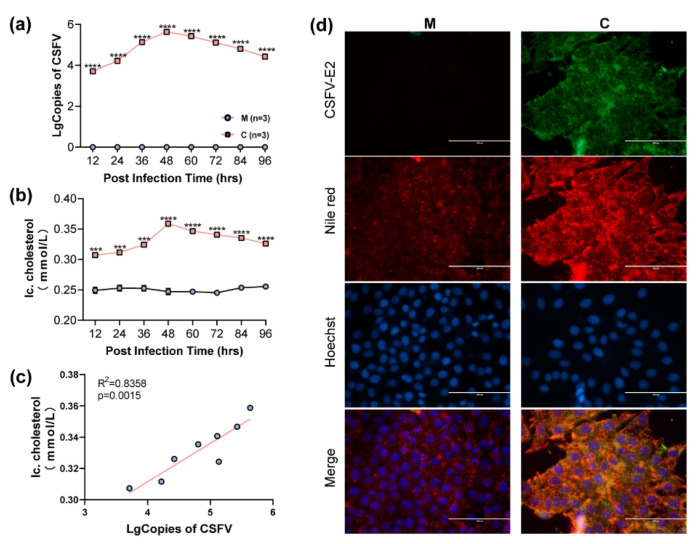
**The alterations of intracellular cholesterol in CSFV-infected PK-15 cells.** (**a**) CSFV one-step growth curve in PK-15 cells; n = 3; **** *p* < 0.0001. (**b**) Variation curve of intracellular cholesterol contents in CSFV-infected PK-15 cells; n = 3; *** *p* < 0.001; **** *p* < 0.0001. (**c**) Correlation analysis of CSFV replication and intracellular cholesterol levels. (**d**) The CSFV E2 protein and intracellular cholesterol levels in CSFV-infected PK-15 cells were assessed by IFA and Nile red staining. **(M, mock; C, CSFV).**

**Figure 2 viruses-14-01450-f002:**
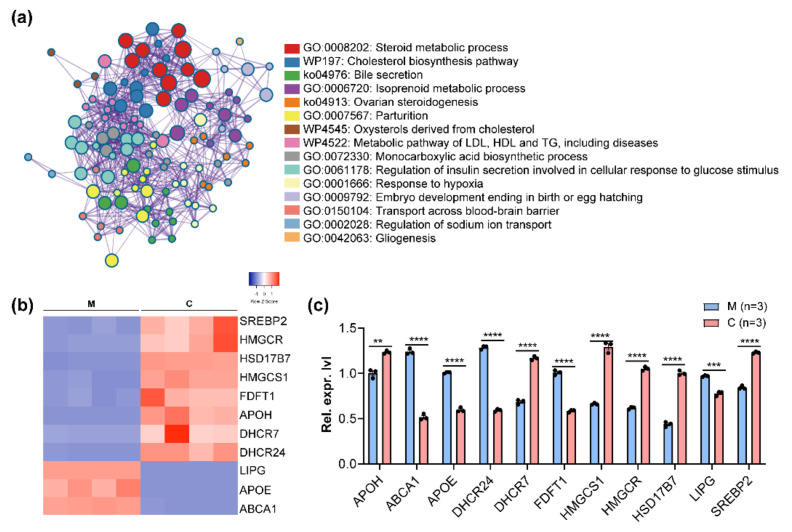
**Cholesterol biosynthesis in CSFV-infected PK-15 cells.** (**a**) GO and KEGG enrichment for differentially expressed genes in CSFV-infected PK-15 cells. (**b**,**c**) The expression of gene involved in cholesterol biosynthesis was detected in CSFV-infected PK-15 cells; ** *p* < 0.01; *** *p* < 0.001; **** *p* < 0.0001. **(M, mock; C, CSFV; Rel. expr. lvl, relative expression level).**

**Figure 3 viruses-14-01450-f003:**
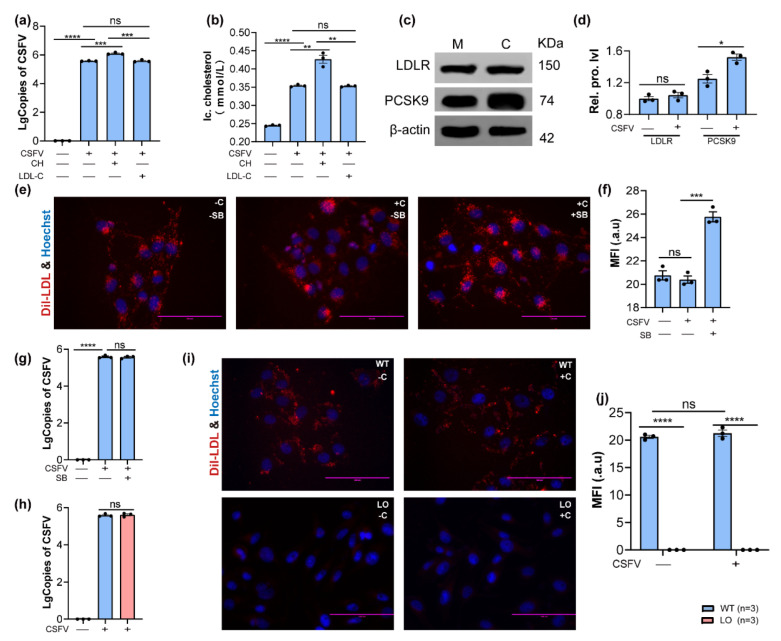
**PCSK9 expression and exogenous cholesterol uptake during CSFV infection.** (**a**) The viral genome copies of CSFV in PK-15 cells at 48 hpi were detected by qPCR; n = 3; *** *p* < 0.001; **** *p* < 0.0001. (**b**) The detection of intracellular cholesterol levels in CSFV-infected PK-15 cells; n = 3; ** *p* < 0.01; **** *p* < 0.0001. (**c**,**d**) The protein-expression levels of LDLR and PCSK9; n = 3; * *p* < 0.05. (**e**,**f**) The detection of dil-LDL uptake in CSFV-infected PK-15 cells at 48 hpi; n = 3; *** *p* < 0.001. (**g**) The inhibition of PCSK9 expression did not affect CSFV replication; n = 3; **** *p* < 0.0001. (**h**) The disruption of LDLR did not affect CSFV replication; n = 3. (**i**,**j**) The uptake of dil-LDL was completely blocked by the deletion of LDLR; n = 3. **(WT, wild-type; LO, ldlr-ko; M, mock; C, CSFV; Rel. prot. lvl, relative protein level; Ic. cholesterol, intracellular cholesterol level; MFI, mean fluorescent intensity).**

**Figure 4 viruses-14-01450-f004:**
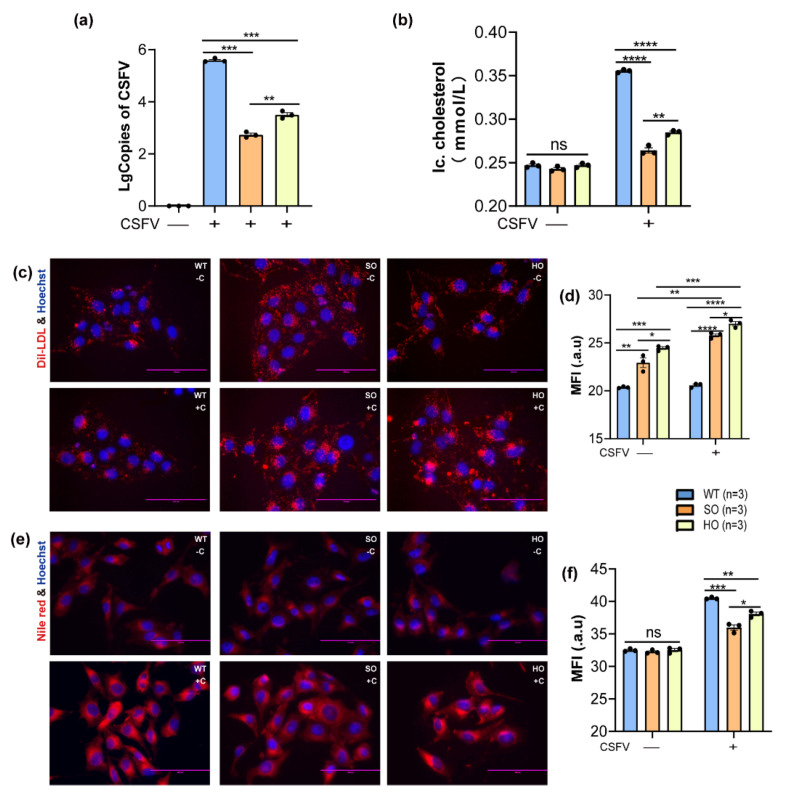
**Cholesterol biosynthesis regulates CSFV replication in PK-15 cells**. (**a**,**b**) The disruption of cholesterol biosynthesis effectively reduced intracellular cholesterol contents and CSFV replication; n = 3; ** *p* < 0.01; *** *p* < 0.001; **** *p* < 0.0001. (**c**,**d**) The deficiency of cholesterol biosynthesis upregulated dil-LDL uptake and decreased intracellular cholesterol levels; n = 3; * *p* < 0.05; ** *p* < 0.01; *** *p* < 0.001; **** *p* < 0.0001. (**e**,**f**) **(WT, wild-type; SO, sqle-ko; HO, hsd17b7-ko; Ic. cholesterol, intracellular cholesterol level; MFI, mean fluorescent intensity).**

**Table 1 viruses-14-01450-t001:** **The primers used in cell clones’ genotyping-PCR.**

Primers	Sequences (5′ to 3′)	Amplicon (bp)
LDLR-JD	TGCATAGCCAGACTCTCTTGG	904
	TGTGATCTCCCATTGCAATCTA	
SQLE-JD	GGTGTCACCGAAGAAGCCTT	819
	GAACCTCCATATGCCGCTGG	
HSD17B7-JD	GGAAGGCTATTGCAAGTGCC	933
	TGGGTCTCACTGACGTCTCT	

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
