# Peer review of "Cholesterol Biosynthesis Modulates CSFV Replication"

_viruses, 2022, doi:10.3390/v14071450_

Round 1

Reviewer 1 Report

In this study researchers looked for the role of cholesterol biosynthesis on CSFV replication using a PK15 cell line. The introduction provides enough background, methods are well described and results and discussion clearly illustrate the data presented. Few comments to improve the manuscript are as follows:

1. In figure 1, what was the response between 0 to 12 hours? Inclusion of that would provide better information about the progression of virus replication and cholesterol biosynthesis. 

2. Data from major CSFV target porcine cells, like macrophages, are missing in this manuscript. 

Reviewer 2 Report

In the manuscript "Cholesterol biosynthesis modulates CSFV replication" sent to me for review the authors describe the results of the research on the correlation between CSFV infection and cholesterol levels in infected cells (PK-15).

The work begins with a brief and logical introduction discussing the economic effects of CSFV infection in pigs as well as presenting typical symptoms of the disease caused by this virus. Then, the authors, based on the literature reports, describe the infection mechanism. It was emphasized that the disturbance of lipid metabolism homeostasis is an important feature of many viral infections, and that viruses use the host's lipid metabolism to support the self-replication cycle and weaken the immune response of the infected organism. Also in the case of SARS-Cov-2 infection, acylation of the proteins of the viral spikes with the fatty acid residues  plays an important role, in order to increase the infectivity of the virus.

As in the case of the previously discussed viruses, also in the case of CSFV infection, the presence of free fatty acids seems to be crucial for the enhancement of viral replication.

Based on the conducted studies, it was shown that the intracellular cholesterol level in PK-15 cells was increased during CSFV infection. Additionally, studies have shown that cholesterol biosynthesis was significantly enhanced in CSFV-infected cells and virus hijacks  host energy metabolism. The presented results seem to be interesting. The entire manuscript is written in a logical and correct manner.

When it comes to minor technical notes, this a space should be used when recording the temperature between the numerical value and the unit (in the manuscript record of temperatures is inconsistent, sometimes there is a space, and sometimes there is no space). In the description of the reagents used for testing, when FBS is mentioned, it is worth adding the information whether the reagent was thermally inactivated before use.

Concerning the summary, the authors only suggest further studies are warranted to elucidate the precise mechanisms between cholesterol biosynthesis and CSFV replication. This leaves a certain insufficiency. However, since the authors decided to publish the results of their research at this stage, it is worth speculating at least on how the information obtained so far could be used to reduce the infectivity of the virus.

Reviewer 3 Report

The paper could be accepted after language revision.

Author Response

Thank you for your suggestions. According to your advice, this manuscript was edited for proper English language, grammar, punctuation, spelling, and overall style by one or more of the highly qualified native English-speaking editors at MDPI.

Round 2

Reviewer 1 Report

Thanks for addressing the comments raised earlier. Still, it will be important to use macrophages or generate in vivo data moving forward.